# Clinical, Pathological and Prognostic Features of Rare BRAF Mutations in Metastatic Colorectal Cancer (mCRC): A Bi-Institutional Retrospective Analysis (REBUS Study)

**DOI:** 10.3390/cancers13092098

**Published:** 2021-04-27

**Authors:** Maria Alessandra Calegari, Lisa Salvatore, Brunella Di Stefano, Michele Basso, Armando Orlandi, Alessandra Boccaccino, Fiorella Lombardo, Alessandra Auriemma, Ina Valeria Zurlo, Maria Bensi, Floriana Camarda, Marta Ribelli, Raffaella Vivolo, Alessandra Cocomazzi, Carmelo Pozzo, Michele Milella, Maurizio Martini, Emilio Bria, Giampaolo Tortora

**Affiliations:** 1Comprehensive Cancer Center, Fondazione Policlinico Universitario Agostino Gemelli IRCCS, Largo Agostino Gemelli 8, 00168 Rome, Italy; mariaalessandra.calegari@guest.policlinicogemelli.it (M.A.C.); brunella.distefano@guest.policlinicogemelli.it (B.D.S.); michele.basso@policlinicogemelli.it (M.B.); armando.orlandi@policlinicogemelli.it (A.O.); maria.bensi01@icatt.it (M.B.); floriana.camarda01@icatt.it (F.C.); marta.ribelli01@icatt.it (M.R.); raffaella.vivolo@unicatt.it (R.V.); carmelo.pozzo@policlinicogemelli.it (C.P.); emilio.bria@policlinicogemelli.it (E.B.); giampaolo.tortora@policlinicogemelli.it (G.T.); 2Medical Oncology, Università Cattolica del Sacro Cuore, Largo Agostino Gemelli 8, 00168 Rome, Italy; a.boccaccino@studenti.unipi.it (A.B.); inavaleria.zurlo@unicatt.it (I.V.Z.); 3Department of Translational Research and New Technologies in Medicine and Surgery, University of Pisa, 56126 Pisa, Italy; 4Section of Oncology, Department of Medicine, University of Verona School of Medicine and Verona University Hospital Trust (AOUI Verona), P.le L.A. Scuro 10, 37134 Verona, Italy; filombardo@ospedalepederzoli.it (F.L.); alessandra.auriemma@aovr.veneto.it (A.A.); michele.milella@univr.it (M.M.); 5Anatomia Patologica, Fondazione Policlinico Universitario “Agostino Gemelli” IRCCS, Largo Agostino Gemelli 8, 00168 Rome, Italy; alessandra.cocomazzi@libero.it (A.C.); maurizio.martini@unicatt.it (M.M.)

**Keywords:** metastatic colorectal cancer, molecular profile, BRAF, rare mutation, V600E, non-V600

## Abstract

**Simple Summary:**

Somatic *BRAF* mutations occur in approximately 10% of metastatic colorectal cancers (mCRCs) and, according to the involved codon, are classified as V600E and in non-V600, accounting for 80% and 20%, respectively. Being the most frequent mutation, the *BRAF* V600E mutation has been extensively investigated and up to now its clinical, pathological and molecular phenotype and its prognostic impact have been clearly described. On the contrary, evidence concerning *BRAF* non-V600 is weaker. We retrospectively evaluated 537 mCRC patients treated at two Italian Institutions. This study corroborates and strengthens available evidence concerning phenotype and prognostic performance of *BRAF* non-V600 compared to *BRAF* V600E and *BRAF* wild-type mCRCs. This deeper insight on rare *BRAF* non-V600 mutated mCRC is a primary issue in the precision oncology era, since the wider application of NGS is expected to increase the identification of those aberrations.

**Abstract:**

Recently, retrospective analysis began to shed light on metastatic colorectal cancers (mCRCs) harboring rare *BRAF* non-V600 mutations, documenting a distinct phenotype and a favorable prognosis. This study aimed to confirm features and prognosis of rare *BRAF* non-V600 mCRCs compared to *BRAF* V600E and *BRAF* wild-type mCRCs treated at two Italian Institutions. Overall, 537 cases were retrospectively evaluated: 221 *RAS/BRAF* wild-type, 261 *RAS* mutated, 46 *BRAF* V600E and 9 *BRAF* non-V600. Compared to *BRAF* V600E mCRC, *BRAF* non-V600 mCRC were more frequently left-sided, had a lower tumor burden and displayed a lower grade and an MMR proficient/MSS status. In addition, non-V600 mCRC patients underwent more frequently to resection of metastases with radical intent. Median overall survival (mOS) was significantly longer in the non-V600 compared to the V600E group. At multivariate analysis, only age < 65 years and ECOG PS 0 were identified as independent predictors of better OS. *BRAF* V600E mCRCs showed a statistically significant worse mOS when compared to *BRAF* wild-type mCRCs, whereas no significant difference was observed between *BRAF* non-V600 and *BRAF* wild-type mCRCs. Our study corroborates available evidence concerning incidence, clinicopathologic characteristics and prognosis of *BRAF*-mutated mCRCs.

## 1. Introduction

Somatic *BRAF* mutations occur in approximately 10% of metastatic colorectal cancers (mCRCs) [1,2].

Of those, V600E missense mutation (*BRAF* V600E) represents the most frequent (accounting for around 80% [3]), being therefore the most investigated. *BRAF* V600E identifies a subgroup of mCRCs with specific clinical (elder age, female gender, right-sided primary tumor location, pattern of metastasization to lymph nodes and peritoneum), pathological (mucinous histology and poor differentiation) and molecular features (CpG island methylator phenotype and high microsatellite instability/mismatch repair deficiency) [2,4,5,6,7,8]. Moreover, *BRAF* V600E displays a well-described negative prognostic impact [9,10,11,12,13,14,15,16]. On the contrary, its predictive role toward anti-EGFRs, either as monotherapy or in combination with chemotherapy, is still debated and some evidence leans toward a negative role [9,17,18]. In mCRC setting, single-agent anti-BRAF target therapy did not show efficacy, due to the ability of neoplastic cells to activate alternative pathways in response to the sole BRAF inhibition [19]. Recent evidence shows that *BRAF* V600E can be effectively targeted by a combination of anti-EGFR and anti-BRAF agents [20,21,22].

Besides the more frequent *BRAF* V600E, mutations occurring outside of codon 600 (*BRAF* non-V600) account for around 20% [3] and have been less analyzed. Recent retrospective analysis shed light on mCRCs harboring *BRAF* non-V600, documenting a distinct phenotype (arising more frequently in younger and male patients and being mainly left-sided, low-grade, non-mucinous, with no peritoneal spread and microsatellite stable) and a favorable prognosis (with a significantly longer survival compared with *BRAF* V600E-mutated mCRCs) [3,23].

Lately, BRAF mutants have been classified into 3 functional classes based on the outcome of mutation on kinase activity: class 1 (BRAF V600) and class 2 are constitutively activated and RAS-independent, whereas class 3 mutants have impaired kinase activity or are kinase-dead, being therefore sensitive to inhibition of activated RAS [24]. A recent analyses of *BRAF*-mutated mCRCs based on functional classes showed that class 3 only displays a divergent phenotype and a better prognosis compared to class 1, while class 2 behaves similarly to class 1 both in terms of features and prognosis [25].

The main limit of evidence concerning *BRAF* non-V600 mutations depends on the rareness of such aberrations. Thus, with the purpose to strengthen such evidence, we conducted a retrospective analysis aiming to confirm the features and prognostic performance of rare *BRAF* non-V600-mutated mCRCs compared to *BRAF* V600E-mutated mCRCs treated at two Italian Institutions. In addition, prognostic performances of *BRAF*-mutated mCRCs (either V600E or non-V600) were compared to *BRAF* wild-type datasets.

## 2. Materials and Methods

This is an observational, retrospective study conducted at two Italian Institutions: Fondazione Policlinico Universitario “Agostino Gemelli” IRCCS, Roma, Italy, and Azienda Ospedaliera Universitaria Integrata, Verona, Italy. The study was approved by the Ethics Committee of Fondazione Policlinico Universitario “Agostino Gemelli” IRCCS, (Roma, Italy).

The primary objective of the study was to assess and compare demographic, clinical, pathological and molecular features of mCRCs harboring rare *BRAF* non-V600 mutations to those of *BRAF* V600E-mutated mCRCs. Secondary objectives were to estimate the prognostic performance of *BRAF* non-V600 compared to *BRAF* V600E in mCRCs, and to compare the prognostic performance of *BRAF* V600E- and *BRAF* non-V600-mutated mCRCs to that of two additional datasets of *BRAF* wild-type mCRCs (respectively *RAS*-mutated and *RAS* wild-type) treated at Fondazione Policlinico Universitario “Agostino Gemelli” IRCCS, Roma, Italy, during the same time period.

Clinical charts of patients affected by mCRCs treated between January 2013 and December 2018 were reviewed. Patients with tumors harboring *BRAF* missense mutations, treated with at least one line of chemotherapy and evaluable for survival, were enrolled. Baseline demographic and clinical characteristics, as well as information concerning treatments and survival, were collected from clinical charts. Pathological and molecular features were retrieved from histological reports. The following data were collected: gender, age and ECOG PS at diagnosis of metastatic disease, onset of metastatic disease, primary tumor location, sites of metastases, mucinous histology, grade of differentiation, *BRAF* missense mutation, *RAS* mutational status and MSI/MMR status. Moreover, information concerning received treatments and date of death were retrieved. *RAS* and *BRAF* gene mutational status was assessed by means of NGS or pyrosequencing on formalin-fixed, paraffin-embedded (FFPE) archival tumor tissue samples from primary tumor or metastases. Expression of MMR proteins was performed by immunohistochemistry: expression of MLH1, MSH2, MSH6 and PMS2 was tested.

### 2.1. DNA Extraction and Assessment of RAS and BRAF Mutational Status

DNA was extracted from three 10 μm-slides from FFPE tumor tissue samples using the QIAamp DNA FFPE Tissue Kit (QIAGEN, Milan, Italy), following the manufacturer’s protocol. In order to minimize contamination by normal cells, the tumor areas dissected for DNA extraction contained at least 70% of tumor cells. *KRAS*, *NRAS* and *BRAF* mutational analyses were carried out by pyrosequencing on chemo-naïve primary or metastatic samples, as previously described [26]. *RAS* and *BRAF* mutational analysis were performed using the therascreen KRAS Pyro Kit, therascreen RAS Extension Pyro Kit and therascreen BRAF Pyro Kit (Qiagen, Milan, Italy). Mutation status was determined by pyrosequencing on the Qiagen PyroMark Q24.

### 2.2. Assessment of MMR/MSI Status

MMR status was assessed on FFPE tumor tissue samples using antibodies against MLH1, MSH2, MSH6 and PMS2, as previously described [27]. Briefly, FFPE sections (4 μm thick) were mounted on positive charged glass slides. For antigen retrieval, deparaffinized and rehydrated sections were treated with citric acid buffer (pH 6.0), two cycles of 3 min each at 500 W, followed by inhibition of endogenous peroxidase with 3% H_2_O_2_ for 5 min. Then, sections were incubated for 1 h at room temperature with mouse monoclonal anti-MLH-1, anti-MSH2 and anti-MSH6 (clone M1; clone G219-1129; clone 44; Roche, Monza, Italy) and mouse monoclonal anti-human PMS2 (clone 2G5; Novus Biologicals, Milan, Italy). The primary antibodies were visualized using the avidin–biotin–peroxidase complex. Samples were stained more than once, and results were highly reproducible. Staining patterns of MMR proteins were evaluated using normal epithelial, stromal or inflammatory cells, or centers of lymphoid follicles as internal controls. Deficiency of any products of these four MMR proteins was stated as MMR-deficient, while proficient MMR was determined if all MMR proteins were expressed.

MSI analysis was also carried out on all samples. DNA from tumor tissues was analyzed with the Titano MSI kit (Diatech Pharmacogenetics, Jesi, Italy) following the manufacturer’s protocol. MSS was defined if no instability at any of the loci was detected, MSI-low (MSI-L) was defined if instability at a single locus was detected and MSI-high (MSI-H) was defined if two or more loci demonstrated instability.

### 2.3. Statistical Analysis

Endpoint for prognostic assessment was OS. OS was defined as the period from the date of diagnosis of metastatic disease to the date of death due to any cause or was censored at the date of last follow-up for alive patients. Differences in baseline characteristics were compared with the use of the chi-square test. Survivals were estimated according to the Kaplan–Meier method and survival curves were compared using the log-rank test. The correlation of baseline characteristics with OS was assessed in univariate analyses: gender (male vs. female), age and ECOG PS at diagnosis of metastatic disease (<65 vs. >65 years; 0 vs. 1–2), metastases onset (metachronous vs. synchronous), number of metastatic sites (1 vs. >1), primary tumor location (left vs. right), presence of liver, lymph node, lung and peritoneal metastases (no vs. yes), MSI/MMR status (MSI-H/MMR-deficient vs. MSS/MMR-proficient), *BRAF* mutational status (*BRAF* non-V600 vs. *BRAF* V600E), mucinous histology (yes vs. no) and tumor grading (2 vs. 3). Variables with a *p*-value of 0.05 or lower at the univariate analysis were included as covariates at multivariate Cox regression analysis, to examine the effect of baseline prognostic characteristics on OS. Statistical significance was set at *p* = 0.05. All analyses were performed using MedCalc software version 9.2, MedCalc Software Ltd, Ostend, Belgium.

## 3. Results

From January 2013 to December 2018, 55 patients with mCRC harboring *BRAF* missense mutations were treated consecutively at Fondazione Policlinico Universitario “Agostino Gemelli” IRCCS, Roma, Italy, and at Azienda Ospedaliera Universitaria Integrata, Verona, Italy.

Of those, 46 tumors (84%) displayed a V600E mutation (V600E cohort) and 9 (16%) a non-V600 mutation (non-V600 cohort). Within the non-V600 cohort, 3 mutations (K601E, G469A, G469R) belonged to class 2, while 5 mutations (G466E, G466A, two D594G, D594N), belonged to class 3 (Table 1). One patient harbored a T599I mutation, whose kinase activity is unknown. Nevertheless, this mutation is predicted as deleterious (PROVEAN score −5.495; cutoff −2.5) according to PROVEAN web server v1.1.3 [28]. 

### 3.1. Patients’ Characteristics

Patients’ characteristics according to *BRAF* mutational status are shown in Table 2.

Among the *BRAF* V600E cohort, there were 24 (52.2%) female and 22 (47.8%) male patients. Median age at diagnosis of metastatic disease was 66 (ranging from 42 to 85 years), and 24 patients (52.2%) were aged > 65 years. Twenty-two patients (54.3%) had an Eastern Cooperative Oncology Group (ECOG) performance status (PS) of 0. Metastatic disease was diagnosed as synchronous in 27 patients (58.7%). Primary tumor location was right-sided in 25 patients (54.3%), while neoplasms arose from the left-sided colon in 21 cases (45.7%). Disease presented involving multiple metastatic sites in 33 cases (71.7%). Metastases affected liver in 30 patients (65.2%), lymph nodes in 28 patients (60.8%) and lung in 18 patients (39.1%), while peritoneal involvement was reported in 14 cases (30.4%) and bone metastases were diagnosed in 7 cases (15.2%). Out of 46 tumors, 11 had a mucinous histology (23.9%) and 27 were high grade (58.7%). According to the BRAF Be Cool prognostic score, 26 cases (56.5%), 19 (41.3%) and 1 (2.2%) were classified as low, intermediate and high risk, respectively [11]. Mismatch repair (MMR)/microsatellite instability (MSI) status was available for 33 samples (71.7%). Of those, 8 tumors (17.4%) displayed a deficit of MMR proteins’ expression (involved in all cases MLH1 and PMS2) or a high microsatellite instability (MSI-H). No case of concomitant *RAS* mutation was reported. Eight patients (17.4%) received surgery for metastases with radical intent. Five patients (10.8%) received an anti-EGFR based treatment; of those, four patients (80%) experienced a disease progression as best response.

The *BRAF* non-V600E cohort consisted of 6 males and 3 females. Age at diagnosis ranged from 45 to 79 years, median age was 61 years and 5 patients were aged < 65 years (55.5%). Five patients had an ECOG PS of 0 (55.5%). Metastatic disease was diagnosed as synchronous in 6 patients (66.6%). Primary tumor location involved the left colon in 8 patients (88.9%), while 1 tumor arose from the right colon. Disease involved single rather than multiple metastatic sites (66.6% vs. 33.4%). Metastatic spread affected liver in 7 cases (77.8%), while lymph nodes were involved in 3 cases (33.3%), peritoneum in 2 cases (22.2%) and lung and bone in 1 case each (11.1%). Only one tumor had a mucinous histology (11.1%), whereas 7 out 9 cases were moderately differentiated (77.7%). MMR status was available for 7 (77.7%) cases, and all tumors displayed an MMR-proficient/microsatellite stable (MSS) status. All tumors were *RAS* wild-type. Six patients underwent surgery for metastases (66.6%). In two patients (22.2%), an anti-EGFR-based treatment was delivered, achieving in both cases (100%) a partial response as best response. Of those, 1 case belonged to class 2 and 1 to class 3.

Compared to *BRAF* V600E mCRCs, *BRAF* non-V600 mCRCs were more frequently left-sided (*p* = 0.017) and displayed a lower grade (*p* = 0.045) and a MMR-proficient/MSS status (*p* < 0.001). In addition, non-V600 mCRC patients had a lower tumor burden (involving a single metastatic site) (*p* = 0.026). Notably, non-V600 mCRC patients underwent more frequently to resection of metastases with radical intent (66.6 vs. 17.4%; *p* = 0.002).

### 3.2. BRAF Prognostic Performance

At a median FU of 13 months, 25 death events were reported in the V600E cohort. Median FU for the non-V600 cohort was 37 months and 4 death events were reported.

Median overall survival (mOS) was 61.3 months for the *BRAF* non-V600 cohort and 20.4 months for the *BRAF* V600E cohort (Figure 1). mOS was significantly longer in the non-V600 compared to the V600E cohort (61.3 vs. 20.4 months; HR 0.41, 95%CI 0.18–0.93; *p* = 0.05) (Figure 1).

Additionally, the prognostic performance of *BRAF* V600E- and *BRAF* non-V600-mutated mCRCs was compared to that of two additional datasets of *BRAF* wild-type mCRCs (respectively *RAS*-mutated and *RAS* wild-type) treated at Fondazione Policlinico A. Gemelli-IRCCS during the same time period. Overall, a total of 537 mCRCs were retrospectively evaluated: 221 *RAS/BRAF* wild-type (41%), 261 *RAS*-mutated (48.5%), 46 *BRAF* V600E (8.5%) and 9 *BRAF* non-V600 (2%). Median OS was 61.3 months for patients with *BRAF* non-V600 mCRCs, 20.4 months for *BRAF* V600E mCRCs, 36.1 months for *RAS/BRAF* wild-type mCRCs and 30.3 months for *RAS*-mutated mCRCs (log-rank test *p* = 0.0066) (Figure 2).

*BRAF* V600E mCRCs showed a statistically significant worse mOS when compared to *BRAF* wild-type mCRCs, for both *RAS* wild-type (20.4 vs. 36.1 months; HR 1.91, 95%CI 1.1–3.29; *p* = 0.002) (Appendix A) and *RAS*-mutated (20.4 vs. 30.3 months; HR 1.59, 95%CI 0.96–2.64; *p* = 0.023) (Appendix A). No statistically significant difference in terms of mOS was observed between *BRAF* non-V600 mCRCs and *BRAF* wild-type mCRCs, for both *RAS* wild-type (61.3 vs. 36.1 months; HR 0.67, 95%CI 0.29–1.54; *p* = 0.43) (Appendix A) and *RAS*-mutated (61.3 vs. 30.3 months; HR 0.53, 95%CI 0.25–1.11; *p* = 0.2) (Appendix A).

In the univariate analysis, besides *BRAF* non-V600 mutation (HR 0.41; 95%CI 0.18–0.93; *p* = 0.05), the following baseline characteristics were associated with increased OS: age < 65 years (HR 03.5; 95%CI 0.16–0.74; *p* = 0.0045), ECOG PS < 1 (HR 0.31; 95%CI 0.13–0.7; *p* = 0.0007), single site of metastatic spread (HR 0.29; 95%CI 0.14–0.61; *p* = 0.0029), absence of lung metastases (HR 0.45; 95%CI 0.19–1.03; *p* = 0.02) and G2 tumor grading (HR 0.26; 95%CI 0.12–0.54; *p* = 0.0004) (Table 3). At multivariate analysis, only age < 65 years (HR 0.38; 95%CI 0.16–0.92; *p* = 0.034) and ECOG PS < 1 (HR 0.23; 95%CI 0.09–0.59; *p* = 0.002) retained their prognostic impact on OS, being identified as independent predictors of better OS (Table 3).

## 4. Discussion

Our study corroborates available evidence concerning incidence and clinical, pathological and molecular phenotypes of *BRAF*-mutated mCRCs (both V600E and non-V600).

In our study population, non-V600 mutations accounted for 20% of *BRAF* mutations. Considering the whole cohort (including both *BRAF*-mutated and *BRAF* wild-type patients), *BRAF* non-V600 mutations were reported in 2% of cases. These results are in line with previous reports by Jones et al. [3].

Besides incidence, the clinical-pathological phenotype of both *BRAF* V600E- and *BRAF* non-V600-mutated mCRCs reported in our analysis was consistent with available evidence. Notably, *BRAF* V600E mCRCs occurred more frequently in female and elderly patients, arising mainly from the right colon and being associated with high tumor burden (involving multiple rather than single metastatic sites) and with high differentiation grading. On the other hand, *BRAF* non-V600 mCRCs developed mainly in male and younger patients, arising more frequently from the left colon and being associated to a low tumor burden (with a single metastatic site, involving mainly the liver) and a low tumor grading. Comparing patients’ and tumors’ features of V600E and non-V600 cohorts, differences in demographic features (gender and age) did not achieve statistical significance, while left primary tumor location, lower tumor burden and lower tumor grading resulted significantly more frequent in the non-V600 cohort (*p* = 0.017, 0.026 and 0.045, respectively). Moreover, non-V600-mutated mCRCs displayed more frequently an MSS/MMR-proficient status (*p* = 0.00016). In 2015, Cremolini et al. [23] showed that *BRAF* 594/596-mutated mCRCs arose mainly from the rectum, did not develop peritoneal spread and displayed more frequently a non-mucinous histology and a MSS status. In a large, multicenter, retrospective cohort study published by Jones et al. in 2017 [3], *BRAF* non-V600-mutated mCRCs, as compared to *BRAF* V600E-mutated, arose more frequently in younger and male patients, displayed mainly a low grade of differentiation and primary tumor location less frequently involved the right side. More recently, Schirripa et al. [25] analyzed clinical, pathological and molecular phenotypes of *BRAF*-mutated mCRC, according to functional classes based on kinase activity [24]. The analysis reported distinguished features for mCRCs belonging to class 3 and class 2. Indeed, class 2 mutated mCRCs did not diverge from class 1, whereas class 3 mCRCs were more frequently left-sided, with rare lymph node and peritoneal involvement. Moreover, in the analysis by Schirripa et al. [25], class 3 mutated mCRCs did not display any concomitant *RAS* mutation. The model of class 3 BRAF mutants by Yao et al. [24] postulates that signaling dysregulation by these mutants relies on concomitant mechanisms to support RAS activation: coexisting *RAS* mutations or *NF1* deletions or mutations in melanomas, and RAS activation by aberrant expression or activity of receptor tyrosine kinase in epithelial tumors (lung or CRC). In our analysis, no coexisting *RAS* mutation was described for class 3 mutants. Therefore, our results are consistent with data by Yao et al. [24] and Schirripa et al. [25] regarding the possible role of receptor tyrosine kinase aberrant expression or activity in supporting RAS activation in class 3 BRAF mutants. Wider genomic profiling would allow deeper insights to address this issue.

Looking at the treatments administered in our patients’ population, patients belonging to the non-V600 cohort underwent more frequently to surgical resection of metastases compared to those belonging to the V600E cohort (66.6% vs. 17.4%; *p* = 0.001912). Interestingly, this data is consistent with the previous retrospective analysis by Cremolini et al., which documented a higher rate of radical resections in patients with *BRAF* 594/596-mutated mCRCs compared with *BRAF* V600E mCRCs (60% vs. 12%; *p* = 0.001) [23]. Anti-EGFR-based treatments were administered in 5 (10.5%) V600E-mutated mCRCs and in 2 (22.2%) non-V600-mutated mCRCs. Best response was disease progression in 80% of the V600E cohort and partial response in 100% of the non-V600 cohort. No difference in activity and efficacy of anti-EGFR agents was observed between class 2 and 3. However, given the small sample size, clear conclusions cannot be drawn.

Concerning survival analysis, mOS was significantly longer in the non-V600 cohort compared to the V600E cohort (61.3 vs. 20.4 months, HR 0.41, 95% CI 0.18–0.93; *p* = 0.05). We must highlight that the mOS for the *BRAF* V600E cohort observed in our analysis reached 20.4 months, overcoming historical reports of an average of 12 months [9,10,11,12]. This might be due to the prevalence of cases (56.5%) belonging to the low-risk category according to the BRAF Be Cool score [11]. This class has been associated to a less grim prognosis, with mOS reaching up to 30 months. In spite of this, our analysis confirmed the expected negative prognostic impact of *BRAF* V600E mutation compared to *BRAF* wild-type datasets. Indeed, mOS of the *BRAF* V600E cohort was significantly shorter compared to *BRAF* wild-type mCRCs, both *RAS* wild-type and *RAS*-mutated (*p =* 0.002 and *p* = 0.023, respectively). Finally, we should highlight the representativity and the wide sample size of our *BRAF* wild-type datasets compared to those of previous analyses. Indeed, the paper by Jones et al. (which is the most comprehensive analysis published up to now) reported survival data only for 249 of the total of 9435 patients included in the analysis (2.6%) [3].

We can hypothesize that the prognostic value of *BRAF* non-V600 mutations could depend on the different distribution of clinical-pathological features between the two cohorts. Indeed, tumors belonging to the non-V600 cohort displayed a statistically significant prevalence of features which are usually associated with good prognosis in the mCRC setting. Primary location was more frequently left-sided (*p* = 0.017), tumors displayed mainly a low tumor grade (*p* = 0.045) and presented with a low tumor burden (involving a single metastatic site) (*p* = 0.026). As a result of limited metastatic disease involving mainly the liver, a high proportion (66.6%) of patients belonging to the non-V600 cohort underwent liver surgery. This unbalance in clinical-pathological features and surgery with radical intent might have impacted on survival. Univariate and multivariate analyses were carried out in order to address this issue. For univariate analyses, besides *BRAF* non-V600 mutations, other factors positively impacted on survival: age < 65 years, ECOG PS of 0, low tumor burden, absence of lung metastases and low tumor grading. Notably, in accordance with previous evidence, primary tumor location did not appear to have a prognostic value in *BRAF*-mutated mCRC [11]. For multivariate analyses, only ECOG PS of 0 and age < 65 years retained a positive prognostic impact. Interestingly, ECOG PS is the main prognostic factor included in the simplified BRAF Be Cool score [11].

Up to now, *BRAF* mutational status has been usually assessed by means of RT-PCR and pyrosequencing. These techniques enable an assessment limited to exons 15 (codons 599, 600 and 601) and 11 (codons 464–469). During the last years, the wide application of diagnostic next-generation sequencing (NGS) allowed a more extensive analysis [29], increasing the identification of *BRAF* non-V600 missense mutations which would have otherwise been underdiagnosed [3]. Within *BRAF* missense mutations, a role as a poor prognostic factor and target for molecular treatment [20] pertains only to V600E mutation. Thus, assessment of *BRAF* mutational status could involve exclusively codon 600 instead of requiring an extensive analysis. Nevertheless, it is important to know the clinical-pathological, prognostic and predictive features of rare non-V600 mutated mCRCs, whose identification is expected to progressively increase, due to the wider application of NGS.

The main limitation of our study relies on both the retrospective nature and on the small sample size, which might have caused selection biases. Thus, prospective confirmation is warranted. Nevertheless, our results are consistent with previous evidence and allow to corroborate and strengthen data available up to now [3,23,25]. Moreover, given the low incidence of *BRAF* mutations in the setting of mCRC (around 10%) and the even lower incidence of non-V600 mutations (around 2%), it would be difficult to design and carry out prospective analysis to investigate this specific subgroup. Indeed, as previously hypothesized [30], it can be postulated that in the precision oncology era, retrospective analysis might represent a reliable and effective tool to allow an assessment of the prognostic value of rare molecular signatures. In order to limit the biases connected with the retrospective nature and the small sample size, a collaboration within national and international consortia is deemed mandatory and correlations with preclinical data are recommended.

## 5. Conclusions

Our results provide a deeper insight on *BRAF*-mutated mCRC, enlarging evidence on clinical, pathological and prognostic features of this rare entity. Specifically, our results corroborate two hot topics:*BRAF* V600E mutation is associated with poor prognosis in the setting of mCRC.Prognosis of *BRAF*-mutated mCRC significantly varies depending on the specific missense mutation. Namely, rare non-V600 mutations identify a distinct subgroup of patients with favorable prognosis and peculiar phenotype.

These findings highlight the need for routine evaluation of *BRAF* mutational status and furthermore, for determining the specific missense mutation, in order to guide clinicians in decision making.

## Figures and Tables

**Figure 1 cancers-13-02098-f001:**
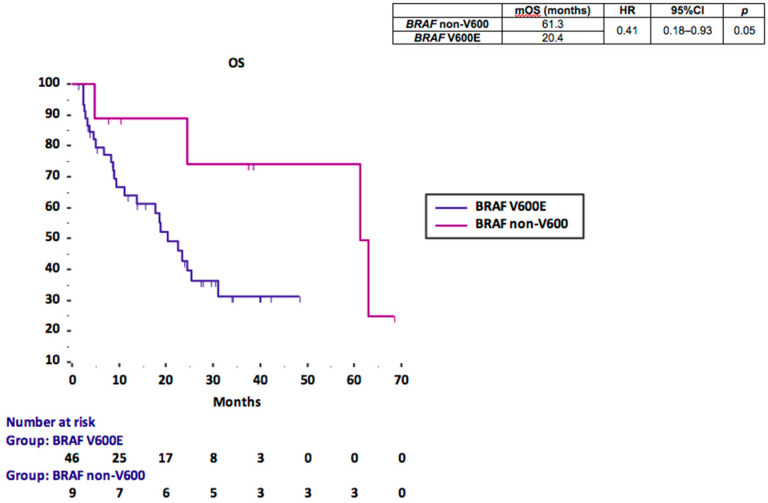
Kaplan–Meier OS curves according to *BRAF* mutational status.

**Figure 2 cancers-13-02098-f002:**
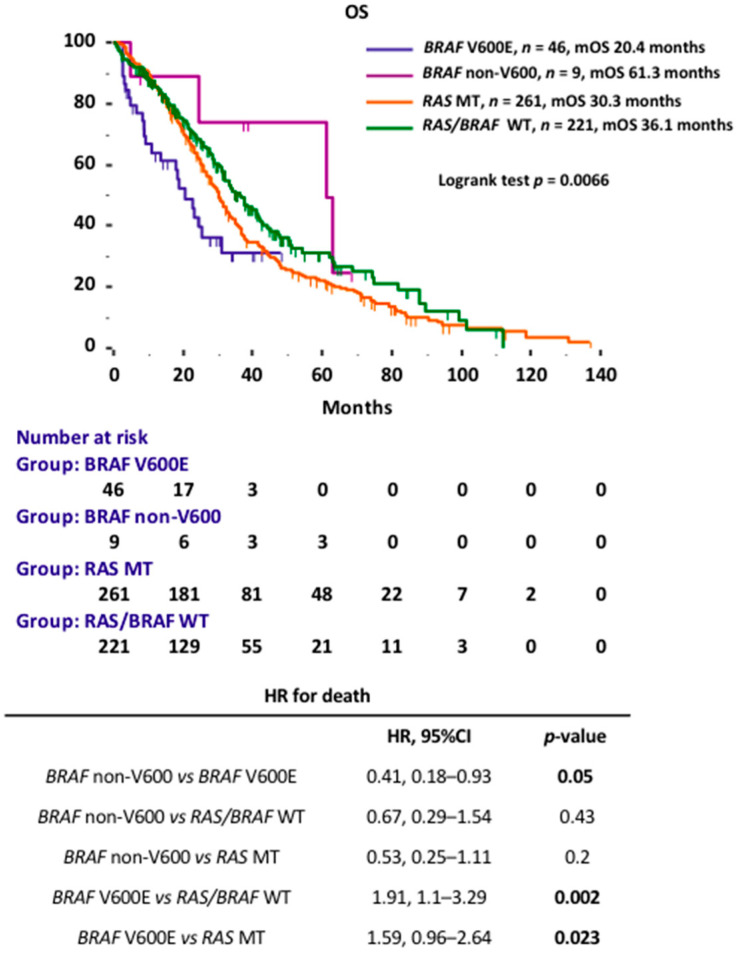
Kaplan–Meier OS curves according to *RAS/BRAF* mutational status.

**Table 1 cancers-13-02098-t001:** Classification of *BRAF* mutations in patients’ population.

*BRAF* Mutation	Class	Kinase Activity	Number of Patients, *N* (%)
V600E	1	High	46 (84)
K601E	2	High	1 (1.8)
G469A	2	High	1 (1.8)
G469R	2	Intermediate	1 (1.8)
G466E	3	Low	1 (1.8)
G466A	3	Low	1 (1.8)
D594N	3	None	1 (1.8)
D594G	3	None	2 (3.4)
T599I	/	Unknown	1 (1.8)

**Table 2 cancers-13-02098-t002:** Patients’ characteristics according to *BRAF* mutational status.

Characteristics	*BRAF* V600E Total = 46, *N* (%)	*BRAF* Non-V600 Total = 9, *N* (%)	*p*
Gender			
Female	24 (52.2)	3 (33.4)	0.3
Male	22 (47.8)	6 (66.6)	
Age			
median, years	66	61	
range, years	42–85	45–79	
≤65 years	22 (47.8)	5 (55.5)	0.6
>65 years	24 (52.2)	4 (44.5)	
ECOG PS			
0	25 (54.3)	5 (55.5)	0.9
1–2	21 (45.7)	4 (44.5)	
Metastases onset			
Synchronous	27 (58.7)	6 (66.6)	0.6
Metachronous	19 (41.3)	3 (33.4)	
Mucinous histology			
Y	11 (23.9)	1 (11.1)	0.39
N	35 (76.1)	8 (88.9)	
Primary tumor location			
Right	25 (54.3)	1 (11.1)	**0.017**
Left	21 (45.7)	8 (88.9)	
Number of metastatic sites			
1	13 (28.3)	6 (66.6)	**0.026**
>1	33 (71.7)	3 (33.4)	
Metastatic sites			
Liver metastases	30 (65.2)	7 (77.8)	0.46
Lymph node metastases	28 (60.8)	3 (33.4)	0.12
Lung metastases	18 (39.1)	1 (11.1)	0.1
Peritoneal metastases	14 (30.4)	2 (22.2)	0.61
Grading			
2	19 (41.3)	7 (77.8)	**0.045**
3	27 (58.7)	2 (22.3)	
MSI/MMR status			
MSI-H/MMR-deficient	8 (17.4)	/	**<0.001**
MSS/MMR-proficient	25 (54.3)	7 (77.7)	
NE	13 (28.3)	2 (22.2)	
Resection of metastases			
Y	8 (17.4)	6 (66.6)	**0.002**
N	38 (82.6)	3 (33.4)	
BRAF Be Cool score			
Low risk	26 (56.5)	NA	
Intermediate risk	19 (41.3)	NA	
High risk	1 (2.2)	NA	
Lines of treatment			
1	27 (58.7)	3 (33.4)	0.16
2	11 (23.9)	1 (11.1)	0.39
≥3	8 (17.4)	5 (55.5)	**0.01**
First line regimen			
Triplet-based	7 (15.2)	1 (11.1)	0.74
Doublet-based	39 (84.8)	8 (88.9)	
Administered drugs			
Fluoropyrimidines	45 (97.8)	9 (100)	0.65
Oxaliplatin	42 (91.3)	7 (77.8)	0.23
Irinotecan	22 (47.8)	4 (44.5)	0.85
Raltitrexed	1 (2.2)	/	0.65
Bevacizumab	29 (63)	6 (66.6)	0.83
Aflibercept	1 (2.2)	1 (11.1)	0.19
Cetuximab	4 (8.7)	1 (11.1)	0.81
Panitumumab	1 (2.2)	1 (11.1)	0.19
Anti-BRAF	4 (8.7)	/	0.35
Regorafenib	3 (6.5)	/	0.43
TAS-102	1 (2.2)	2 (22.2)	**0.01**

*p*-values below 0.05 are reported in bold. ECOG, Eastern Cooperative Oncology Group; MMR, mismatch repair; MSI, microsatellite instability; MSI-H, microsatellite instability-high; MSS, microsatellite stable; NA, not applicable; NE, not evaluated; N, no; OS, overall survival; PS, performance status; Y, yes.

**Table 3 cancers-13-02098-t003:** Univariate and multivariate analysis for OS.

Variable	Overall Survival (OS)					
	Univariate Analysis			Multivariate Analysis		
	HR	95%CI	*p*	HR	95%CI	*p*
Gender						
Male vs. Female	0.55	0.25–1.21	0.09			
Age (years)						
<65 vs. >65	0.35	0.16–0.74	**0.004**	0.38	0.16–0.92	**0.034**
ECOG PS						
0 vs. 1–2	0.31	0.13–0.7	**<0.001**	0.23	0.09–0.59	**0.002**
Metastases onset						
Metachronous vs. synchronous	0.86	0.41–1.78	0.69			
Number of metastatic sites						
1 vs. >1	0.29	0.14–0.61	**0.003**	0.28	0.08–1.01	0.052
Primary tumor location						
Left vs. right	0.51	0.23–1.11	0.06			
Liver Metastases						
No vs. Yes	0.74	0.34–1.59	0.46			
Lymph node metastases						
No vs. Yes	0.53	0.25–1.12	0.08			
Peritoneal metastases						
No vs. Yes	0.87	0.37–2.02	0.73			
Lung metastases						
No vs. Yes	0.45	0.19–1.03	**0.02**	0.57	0.24–1.3	0.18
MSI/MMR status						
MSI-H/MMRdef vs. MSS/MMRpro	0.74	0.24–2.29	0.63			
*BRAF*						
non-V600 vs. V600E	0.41	0.18–0.93	**0.05**	0.6	0.12–2.88	0.52
Mucinous histology						
Yes vs. No	0.49	0.21–1.16	0.18			
Grading						
2 vs. 3	0.26	0.12–0.54	**<0.001**	0.5	0.2–1.28	0.15

*p*-values below 0.05 are in bold. CI, confidence interval; ECOG, Eastern Cooperative Oncology Group; HR, hazard ratio; MMR, mismatch repair; MSI, microsatellite instability; MSI-H, microsatellite instability-high; MSS, microsatellite stable; OS, overall survival; PS, performance status.

## Data Availability

Data is contained within the article or Appendix A.

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
