# Peer review of "Clinical, Pathological and Prognostic Features of Rare BRAF Mutations in Metastatic Colorectal Cancer (mCRC): A Bi-Institutional Retrospective Analysis (REBUS Study)"

_cancers, 2021, doi:10.3390/cancers13092098_

Round 1
Reviewer 1 Report
The manuscript “Clinical, pathological and prognostic features of RarE BRAF 2 mUtationS in metastatic colorectal cancer (mCRC): a bi-institu-3 tional retrospective analysis (REBUS study) by Calegari et al is a retrospective analysis on metastatic colorectal cancers (mCRCs) harbouring rare BRAF nonV600 mutations. This study, conducted at two institutional sites, aimed to shed light on the role of rare Braf mutations in mCRC. 537 cases were retrospectively evaluated: 221 RAS/BRAF wild type, 261 RAS mutated, 46 BRAF V600E and 9 BRAF nonV600. According to previous reports BRAF non-V600 mCRC were more frequently left-sided, had a lower tumor burden and displayed a lower grade and a MMR proficient/MSS status.
The study is interesting and well written and the topic is of interest for the scientific community. Nevertheless, few comments can be raised:
- The authors should more clearly state the added value of their work. Is this a confirmatory study?
- Can the author add the patient treatment mainly referred to the 9 BRAF non V600?
- Did any of the patient BRAF non V600 undergo to metastasis genomic profiling?
- Are rare BRAF mutation reported in associated to other mutations?
Author Response
The manuscript “Clinical, pathological and prognostic features of RarE BRAF 2 mUtationS in metastatic colorectal cancer (mCRC): a bi-institu-3 tional retrospective analysis (REBUS study) by Calegari et al is a retrospective analysis on metastatic colorectal cancers (mCRCs) harbouring rare BRAF nonV600 mutations. This study, conducted at two institutional sites, aimed to shed light on the role of rare Braf mutations in mCRC. 537 cases were retrospectively evaluated: 221 RAS/BRAF wild type, 261 RAS mutated, 46 BRAF V600E and 9 BRAF nonV600. According to previous reports BRAF non-V600 mCRC were more frequently left-sided, had a lower tumor burden and displayed a lower grade and a MMR proficient/MSS status.
The study is interesting and well written and the topic is of interest for the scientific community. Nevertheless, few comments can be raised:
- The authors should more clearly state the added value of their work. Is this a confirmatory study? This study aimed to strengthen available evidence on BRAF-mutated mCRCs (which is limited by the rareness of such aberration) by confirming features and prognosis of rare BRAF non-V600 mCRCs compared to BRAF V600E and BRAF wild type mCRCs (page 2 lines 101-102; page 4 lines 148-151). Study results are consistent with and corroborate available evidence concerning incidence, clinicopathologic characteristics and prognosis of BRAF mutated mCRCs (page 2 lines 111-113; page 12 line 347-348).
- Can the author add the patient treatment mainly referred to the 9 BRAF non V600? We previously focused on anti-EGFR treatments (page 12, lines 392-397). Data concerning V600E cohort are reported on page 8 lines 276-279, whereas data concerning non-V600E cohort are reported on page 8 lines 291-293. Following your suggestion, we added information regarding chemotherapy lines (number of treatment lines administered, first line regimen triplet-based vs doublet-based and administered drugs) updating Table 2 (pages 7-8).
- Did any of the patient BRAF non V600 undergo to metastasis genomic profiling? All patients underwent to RAS and BRAF mutational status and MMRP expression assessment. No extensive genomic profiling was performed. Mutational status and MMR status assessment could be performed both on primary tumor or metastatic site. Concordance between both primary and metastatic sites was not assessed. For BRAF nonV600E cohort analysis was performed on primary tumor in 3 cases and on metastatic site in 6 cases.
- Are rare BRAF mutation reported in associated to other mutations? None of the rare BRAF nonV600 mutations was associated with concomitant RAS mutation (stated on page 8, line 290).
Reviewer 2 Report
The study is well written with a lot of descriptive détails of the population. The results are in line with the literature
For me the limits of this study are :
-the retrospective nature of the study
-the small sample size of the target population: 46 BRAFV600E and 9 non V600E
- the not possibility of comparison because it is not a prospective study
-we don’t have the chemotherapy lines treatment for these patients
Author Response
The study is well written with a lot of descriptive détails of the population. The results are in line with the literature
For me the limits of this study are :
-the retrospective nature of the study
-the small sample size of the target population: 46 BRAFV600E and 9 non V600E
- the not possibility of comparison because it is not a prospective study.
We do agree with the reviewer. Indeed, we stated that main limitations of our study rely on both the retrospective nature and on the small sample size, which might have caused selection biases and therefore prospective confirmation is warranted (page 13, lines 441-443). At the same time, our results are consistent with previous evidence and allow to corroborate and strengthen data available up to now. Moreover, given the low incidence of BRAF mutations in the setting of mCRC (around 10%) and the even lower incidence of nonV600 mutations (around 2%), it would be difficult to design and carry out prospective analysis to investigate this specific subgroup.
-we don’t have the chemotherapy lines treatment for these patients. Following reviewer's suggestion, we added information regarding chemotherapy lines (number of lines administered, first line regimen triplet-based vs doublet-based and administered drugs) updating Table 2 (pages 7-8).
Reviewer 3 Report
This is an interesting manuscript investigating the prognostic features of non-V600E BRAF mutants in colorectal cancer, a so far understudied subgroup. As it stands, the manuscript is well-written and the data, incl. the expected small number of non-V600E cases, are well-presented. I have only two minor aspects that, if such data are available or can be obtained in due course, could improve the manuscript:
- in line 272, the authors state that all tumours were RAS wildtype. This is an important finding as four out of the nine on-V600E cases contain class 3 BRAF mutants, which require the presence of activated, not necessarily mutated RAS, to trigger MEK/ERK pathway activation. In the meantime, class 3 BRAF mutant tumours without RAS mutations have been published, e.g. PMID 28783719 and 32913998, but these findings suggest that activated RAS is provided by other means such loss of RAS-GAPs, e.g. NF1, or aberrant expression/activity of receptor tyrosine kinases. Do the authors have data that might identify the alterations that cooperate with the class 3 mutants? If yes, it would be helpful to provide such data. Alternatively, the authors could briefly discuss this aspect as their data show that more comprehensive genetic profiling would provide more insights into the complexities of class 3 BRAF mutant signalling.
- Figure 2. Can the authors spot a difference between class 2 and class 3 mutants in OS. Of course, the numbers are, considering the frequency of the events and a single institution study, too low to make strong claims here, but it would be interesting although at best anecdotal to see whether class 2 and 3 behave similar or whether class 2 mutants are more V600E-like.
Author Response
This is an interesting manuscript investigating the prognostic features of non-V600E BRAF mutants in colorectal cancer, a so far understudied subgroup. As it stands, the manuscript is well-written and the data, incl. the expected small number of non-V600E cases, are well-presented. I have only two minor aspects that, if such data are available or can be obtained in due course, could improve the manuscript:
- in line 272, the authors state that all tumours were RAS wildtype. This is an important finding as four out of the nine on-V600E cases contain class 3 BRAF mutants, which require the presence of activated, not necessarily mutated RAS, to trigger MEK/ERK pathway activation. In the meantime, class 3 BRAF mutant tumours without RAS mutations have been published, e.g. PMID 28783719 and 32913998, but these findings suggest that activated RAS is provided by other means such loss of RAS-GAPs, e.g. NF1, or aberrant expression/activity of receptor tyrosine kinases. Do the authors have data that might identify the alterations that cooperate with the class 3 mutants? If yes, it would be helpful to provide such data. Alternatively, the authors could briefly discuss this aspect as their data show that more comprehensive genetic profiling would provide more insights into the complexities of class 3 BRAF mutant signalling. None of the tumors belonging to class 3 displayed a concomitant RAS mutation. This data is consistent with the report by Schirripa et al (DOI: 10.1158/1078-0432.CCR-19-0311), describing 13 patients with mCRC harboring class 3 BRAF mutations without concomitant RAS mutations. Unfortunately, we do not have data that allow to identify alterations that trigger MEK/ERK pathway activation, cooperating with class 3 mutants. We followed reviewer’s advice to include this aspect in the discussion (page 12, lines 377-386).
- Figure 2. Can the authors spot a difference between class 2 and class 3 mutants in OS. Of course, the numbers are, considering the frequency of the events and a single institution study, too low to make strong claims here, but it would be interesting although at best anecdotal to see whether class 2 and 3 behave similar or whether class 2 mutants are more V600E-like. Median OS was 62.15 months (95%CI 60.97-63.32 months) and 41.09 (95%CI 18.22-63.96 months), respectively for Class 2 (3 cases) and Class 3 (5 cases). Therefore, unlike previous evidence, there was a trend favoring Class 2 over Class 3, although not statistically significant (HR 0.3, 95%CI 0.04-2.27, p=0.28). Compared to Class 1, Class 2 showed a better prognosis (HR 0.21; 95%CI 0.07-0.62) whereas no significant difference was reported between Class 1 and Class 3 (0.64; 95%CI 0.21-1.88). However, numbers are definitely too small to drew reliable conclusions, thus we had decided to exclude this analysis.
Reviewer 4 Report
This retrospective study addresses the important question of the prognostic significance of the rarer NON-V600 BRAF mutation in colorectal cancer. 55 patients from 2 institutions in Italy were analysed. Only 9 patients with NON-V600 BRAF mutation were compared to 46 patients with V600E mutation and found that the prognosis of patients with NON-V600 mutated carcinoma is significantly better and comparable to BRAF wild type.
Major concerns: The basic requirement to establish the clinical value of molecular biology studies is the accurate description of the included population to counteract selection bias. The total number of patients examined is stated only in the "simple summary" section. It is not apparent whether the patients were treated consecutively, which chemotherapy was carried out with which regimen, whether patients in this collective underwent one or even multiple operations in the sense of metastasectomies or other local therapy procedures, etc. As a further indication of a selection bias, it is striking that the patients with the BRAF mutation come from both centres, but those with the wild type only from one. Further indications of a heterogeneous and thus less informative cohort of patients can be found in the distribution and chronological occurrence of the metastases, which show fundamentally different prognoses. Other established prognosis parameters, such as the Fong score, are missing. In other words, the exact description of the cohort of patients examined, including a CONSORT diagram is missing. Therefore, the value of the work is significantly reduced and the present form is not suitable for publication in the journal.
Author Response
This retrospective study addresses the important question of the prognostic significance of the rarer NON-V600 BRAF mutation in colorectal cancer. 55 patients from 2 institutions in Italy were analysed. Only 9 patients with NON-V600 BRAF mutation were compared to 46 patients with V600E mutation and found that the prognosis of patients with NON-V600 mutated carcinoma is significantly better and comparable to BRAF wild type.
Major concerns: The basic requirement to establish the clinical value of molecular biology studies is the accurate description of the included population to counteract selection bias. The total number of patients examined is stated only in the "simple summary" section. It is not apparent whether the patients were treated consecutively, which chemotherapy was carried out with which regimen, whether patients in this collective underwent one or even multiple operations in the sense of metastasectomies or other local therapy procedures, etc. Patients were treated consecutively. We modified the text accordingly, specifying this data at page 6, line 228. We tried to provide the most concise and as deep and exact description as possible of cohorts examined. Nevertheless, information regarding administered treatment lines (with the exception of anti-EGFRs) were missing in the manuscript. Indeed, we had decided not to focus on that since we wanted to address the prognostic (rather than predictive) impact of BRAF mutational status. We followed reviewer’s advice to add information concerning chemotherapy (number of lines administered, first line regimen triplet-based vs doublet-based and administered drugs) updating accordingly Table 2 (pages 7-8). Two patients for each cohort underwent to metastasectomies twice.
As a further indication of a selection bias, it is striking that the patients with the BRAF mutation come from both centres, but those with the wild type only from one. Regarding the concern on the RAS WT cohort, we chose by purpose to limit the wild type cohort to a single institution with the aim of avoiding interlaboratory biases.
Further indications of a heterogeneous and thus less informative cohort of patients can be found in the distribution and chronological occurrence of the metastases, which show fundamentally different prognoses. There was no statistically significant difference between chronological occurrence of metatstases (synchronous vs metachronous). Moreover, while distribution of metastastic sites might reflect different behavior of V600E and nonV600E disease, differences were not statistically significant.
Other established prognosis parameters, such as the Fong score, are missing. Since not all patients underwent to liver metastasectomies, we had decided to report assessment of the BRAF-BeCool score, a more recent, up-to-date and widely applicable score, being useful for patients irrespective of surgeries (data are reported on Table 2, page 7; page 8, lines 271-273; page 13, lines 402-404). Fong score is applicable to patients undergoing liver resection. However, in our V600E cohort less than 20% (N 8) of patients underwent to metastasectomy.
In other words, the exact description of the cohort of patients examined, including a CONSORT diagram is missing. Therefore, the value of the work is significantly reduced and the present form is not suitable for publication in the journal. We did not provide a CONSORT diagram (which is required in case of randomized clinical trial) due to the nature of the study (retrospective, observational).
Round 2
Reviewer 2 Report
The authors'responses are clear and complete with the addition of the required informations.